# Polymer-Derived Carbon Nanofiber and Its Photocurrent-Switching Responses of Carbon Nanofiber/Cu Nanocomposite in Wide Ranges of Excited Light Wavelength

**DOI:** 10.3390/polym15173528

**Published:** 2023-08-24

**Authors:** Xingfa Ma, Mingjun Gao, Xintao Zhang, You Wang, Guang Li

**Affiliations:** 1Center of Advanced Functional Materials, School of Environmental and Material Engineering, Yantai University, Yantai 264005, China; gaomj@ytu.edu.cn (M.G.); zhangxintao@ytu.edu.cn (X.Z.); 2National Laboratory of Industrial Control Technology, Institute of Cyber-Systems and Control, Zhejiang University, Hangzhou 310027, China; king_wy@zju.edu.cn (Y.W.); guangli@zju.edu.cn (G.L.)

**Keywords:** carbon nanofiber, polymer-derived, defect passivation, photocurrent-switching, free carrier extracting and utilization, light/matter interaction

## Abstract

Transformation into electric or photoelectric functional composite from non-conjugated polymers is a great challenge due to the presence of a large number of locative states. In this paper, carbon nanofiber was synthesized via hydrothermal carbonization utilizing carboxymethyl cellulose as a precursor, and the carbon nanofiber/Cu nanocomposite was constructed for defect passivation. The results indicated that the resulting nanocomposites exhibited good absorbance in visible light range and NIR (near-infrared). The photoconductive responses to typical weak visible light (650 nm et al.) and NIR (808, 980, and 1064 nm) were studied based on Au gap electrodes on flexible polymer substrates. The results exhibited that the nanocomposite’s solid thick film showed photocurrent-switching behaviors to visible light and NIR, the switch-ratio was depending on the wavelengths and power of incident lights. The positive and negative photoconductance responses phenomenon was observed in different compositions and changing excited wavelengths. Their photophysical mechanisms were discussed. This illustrated that the nanocomposites easily produce free electrons and holes via low power of incident light. Free electrons and holes could be utilized for different purposes in multi-disciplinary fields. It would be a potential application in broadband flexible photodetectors, artificial vision, simulating retina, and bio-imaging from visible light to NIR. This is a low-cost and green approach to obtain nanocomposite exhibiting good photocurrent response from the visible range to NIR.

## 1. Introduction

Functional nanocomposites have been widely used in new energy [1,2,3], environmental, information, and bio-medical fields [4]. Among so many functional materials, carbon nanostructures are important due to the delocalization of π electrons and conjugated degree. Carbon materials include C_60_, graphene, carbon nanotube, carbon dots, carbon fiber, C_3_N_4_ et al. Carbon dots include graphene quantum dots (GQDs), carbon quantum dots (CQDs), carbon nanodots (CNDs), and carbonized polymer dots (CPDs). Carbon dots are also one of the hot-spot materials because of their good bio-compatibility and fluorescence in the last 10 years. Synthesis of carbon dots mainly involve two kinds of methods: top-down and bottom-up. Materials generally used are precursors containing carbon elements, such as graphite, graphene, carbon tube, carbon fiber, small organic molecular, polymers, biomolecular, and their wastes. Compared to carbon quantum dots, carbonized polymer dots (CPDs) are different because CPDs usually have an incompletely conjugated, sp2/sp3 hybrid carbon skeleton structures. CPDs are generally called a core/shell structured hybrid nanocomposite, i.e., sp2 carbon core, polymer short chain (sp3) as a shell.

It is well known that most of the traditional organic functional materials (conductive, photoelectric, photoluminescence, and nonlinear optical materials) contain π-conjugated polycyclic aromatic structures, and their photophysical performance is often regulated by the conjugation degree of π electrons. For a typical example, increasing or decreasing the π-conjugation length (the size of the π-conjugation ring) or introducing electron-donating/withdrawing groups to tailor the electronic density for the bandgap desired. However, recently, researchers discovered that some non-conjugated and non-aromatic polymers, such as polyamidoamine, maleic anhydride/vinyl acetate alternating copolymers (PMV), polyethylene glycol, polyacrylonitrile, and polyesters, were found to exhibit intrinsic fluorescence at the clustering state. This is mainly the result of aggregation-induced fluorescence. A similar phenomenon was also observed in some natural polymers, including grass, leaf, rice, starch, cellulose, proteins, and so on. These non-conjugated structures do not show fluorescence in an isolated state but show anomalous fluorescence upon aggregation, known as clusteroluminescence (CL) [5]. Due to the tremendously complex aggregation state of materials and materials physics, exploring the relationship between microstructure and properties of materials presents a great challenge. The complex of carbonized polymer dots stems from the primary structure, secondary structure, and aggregation state structure of the polymer. Carbonized polymer dots are considered a core/shell structure, the core is a graphite structure, and its shell is a short polymer chain. This naturally inherits some marks stemming from the precursors (used polymers). Optimizing the physical properties of polymer dots involves bandgap engineering of core materials, and the aggregation state structure of polymer shell. Balancing the delocalization state and locative state of polymer dots is the key to materials design to control the physical and chemical properties. The conditions of dehydration, crosslink, and carbonization have large influences on the graphitization degree of polymer dots, which result in the variation of conductivity, photoelectric, and photoluminescence properties. Based on the mechanism of light excitation and photoluminescence, to suppress non-radiative energy consumption, such as molecular vibration or molecular rotation, the photoluminescence properties would be enhanced greatly. Similarly, to restrain the non-radiative energy consumption, and enhance the graphitization degree of polymer dots or construction of the built-in electrical fields, the photoelectric property would also be increased due to improving the separation of electrons and holes via light inducement. Therefore, the preparation and tailoring of carbon dots showed diverse selection for different applications because of abundant precursors (containing carbon resources). Some representative studies are as follows: Zhang and co-workers [6] prepared carboxymethyl cellulose-derived polymer dots and used it to detect tetracyclines. Bazazi and co-workers [7] introduced polysaccharide-based carbon dots and polysaccharide/carbon-dots nanocomposites. He and co-workers [8] synthesized the cationic polymer-derived carbon dots for enhanced gene delivery and cell imaging. Bhandari and co-workers [9] studied biomolecule-derived quantum dots for sustainable optoelectronics. Sabet and co-workers [10] synthesized nitrogen-doped carbon quantum dots from grass via a simple hydrothermal method, which showed high photoluminescence. Monday and co-workers [11] performed the synthesis and characterization of nitrogen-doped carbon dots from palm kernel shells as precursors. Thangaraj and co-workers [12] reported biomass-derived carbon quantum dots. Li and co-workers [13] prepared silk-derived carbon dots. Lou and co-workers [14] reviewed the progress of biomass carbon dots on syntheses, characterization, luminescence mechanism, and sensing applications. Liu and co-workers [15] reported the indole carbonized polymer dots enhancing full-color emission by tailoring the surface state. Raveendran and co-workers [16] synthesized the mint leaf-derived carbon dots for the detection of Fe(III) and ascorbic acid. Chen and co-workers [17] prepared the graphene quantum dots from natural polymer starch for cell imaging. Yang and co-workers [18] enhanced the fluorescence of polymer carbon dots by cross-linking polyamide chains. Rocco and co-workers [19] synthesized the carbon quantum dots with an electrochemical method. Xia and co-workers [20] introduced carbon dots and carbonized polymer dots. Zhang and co-workers [21] reviewed the natural product-derived carbon dots. Dezfuli and co-workers [22] reported the organic dots for theranostic applications, focused on the preparation and surface engineering of carbon dots. Döring and co-workers [23] introduced chiral carbon dots, focusing on the synthesis, optical properties, and applications. Tao and co-workers [24] enhanced the emission in carbonized polymer dots by confined-domain crosslink. Li and co-workers [25] discussed the fluorescent mechanism of red emissive carbon dots prepared with o-phenylenediamine and catechol. Das and co-workers [26] revealed the nature of optical activity in carbon dots produced from different chiral precursors. Chen and co-workers [27] synthesized the carbon nanodots from organic pollutants and used them for bio-imaging. Guo and co-workers [28] prepared the carbon dots with polythiophene derivatives and discussed their emission mechanism. Sun and co-workers [29] studied photo-thermal and chermodynamic therapy utilizing ferrocene-carbon dot-crosslinked nanoparticles. Gentile and co-workers [30] tailored the chemical structure of nitrogen-doped carbon dots for catalysis applications. Li and co-workers [31] discussed the mechanism of aggregation-induced solid-state fluorescence emission of polythiophene derivatives carbonized polymer dots. Liu and co-workers [32] reported the carbon dots produced with conjugated perylene derivatives, which exhibited strong near-infrared absorption and emission. Wang and co-workers [33] studied the electron–phonon coupling-assisted red luminescence of o-phenylenediamine-based carbon dots. Jiang and co-workers [34] reported hour-level organic long persistent luminescence from carbon dots via covalent fixation. Rafieea and co-workers [35] introduced the modification of carbon-based nanomaterials. Arcudi and co-workers [36] reported the supra-molecular chemistry of carbon-based dots. Stergiou and co-workers [37] studied the interface charge-transfer processes of carbon dots. Tsai and co-workers [38] reported polymer dots as near-infrared fluorescent probes for bio-imaging and sensing. Wang and co-workers [39] studied the solid-state fluorescence of carbon dots and focused on strategies, optical manipulations, and applications. Xu and co-workers [40] reported the solid-state photoluminescence of carbon-based quantum dots. Kang and co-workers [41] introduced aggregation and luminescence in carbonized polymer dots. Ru and co-workers [42] analyzed the aggregation in carbon dots. Feng and co-workers [43] studied the carbon dot/polymer nanocomposites, focused on green synthesis, and applications of energy, environmental and biomedical fields. Shi and co-workers [44] introduced carbon dots as an innovative luminescent nanomaterial. Yu and co-workers [45] reviewed the photo-induced reaction of carbon dots. Pandit and co-workers [46] reported surface-engineered amphiphilic carbon dots. Xu and co-workers [47] introduced the surface functionalization of carbon dots. Wang and co-workers [48] reviewed the applications of carbon dots in bio-imaging, bio-sensing, and therapeutics. Zhang and co-workers [49] introduced multifunctional carbon-based nanomaterials, focused on the applications in biomolecular imaging and therapy. Srinivasan and co-workers [50] synthesized, characterized the nanostructured graphene oxide dots, studied the photoinduced electron transfer, and applied it in the detection of explosives and biomolecules. Zhao and co-workers [51] reported narrow-bandwidth emissive carbon dots and their fluorescent properties. Zheng and co-workers [52] studied the polymer–structure-induced phosphorescence of carbon dots. Hao and co-workers [53] carried out research on carbon dots in applications of memristors. Xue and co-workers [54] reported the formation process and mechanism of carbon dots prepared from aromatic compounds as precursors, and so on.

Reviewed the advances of carbon dots, it is found that most of the studies have focused on photoluminescent properties, analysis, biological imaging, electrocatalysis, photocatalysis, materials physics, and photophysics in the visible light range. Regarding the NIR range, there are a few references. For over ten years, Ma and co-workers [55,56] were interested in organic/inorganic functional nanocomposites and their properties. Especially, free electrons/holes were extracted via light inducement in the broadband spectrum response range (from visible range to NIR). The materials system involved conjugated polymers, metal oxides, metal sulfides, nanodots, carbon dots, conjugated small organic molecules, carbon-based nanocomposites, heterostructures, and organic/inorganic hybrid multifunctional nanocomposites. In previous publications, regarding the studies of polymers, Ma and co-workers mainly focused on the conjugated polymer because of the delocalization of electrons and controlled density of electrons via electron-donating/withdrawing groups or contact interface of nanocomposites [55,57]. In recent years, Ma and co-workers have also been interested in non-conjugated polymer-derived carbon dots or other carbon nanostructures due to the diversified design of nanocomposites. This kind of carbon dots generally contain sp2 and sp3 carbon atoms. The sp2 carbon is advantageous for charge transfer, and sp3 carbon is harmful for carriers extracting. Therefore, the passivation of defects in polymer-derived carbon dots was required to improve the extracting ability of photo-generated carriers. Since metal belongs to free electron-like materials, some typical noble metals, such as Au, Pt, Pd, Ag, etc., are often used for this purpose. Considering the low-cost, Cu nanostructure was selected to passivate the defects of polymer-derived carbon materials in this study. However, Cu generally is easily oxidized. To enhance the oxidation resistance of Cu, and improve its electrical conductivity, Cu-based alloy or nanocomposites were universally accepted, and good effects were obtained. Previous studies indicated that reduced graphene oxide/copper hybrid or copper/silver alloy exhibited improved electrical conductivity and oxidation resistance [58,59,60]. Therefore, polymer-derived carbon/Cu nanocomposite could combine both advantages, exhibiting complementation or synergetic behavior. On one hand, the passivation of defects of polymer-derived carbon was realized with Cu nanostructure for improving the carrier’s extracting ability. On the other hand, interface interaction between polymer-derived carbon and Cu nanostructure contributed to enhancing the oxidation resistance of Cu nanomaterials.

In this paper, to decrease the electronic traps of polymer-derived carbon nanostructure, polymer-derived carbon nanostructure/Cu nanocomposites were prepared for enhancing electron/hole separation and extracting abilities via weak visible (650 nm et al.) and NIR (808, 980, and 1064 nm) light inducement. It would be potential applications in the broadband flexible photodetectors from visible light to NIR, bio-imaging, cancer therapy, artificial vision, remoting drug delivery via light, and simulating retina [61]. Some similar results were also obtained utilizing other carbon nanostructures, such as graphene oxide. Based on the low cost and green synthesis method, these results are also meaningful to interdisciplinary applications and exploring the mechanism of light and matter interaction.

## 2. Experimental Details

### 2.1. Raw Materials

Carboxymethyl cellulose (Industrial grade), Shandong Sanyi New Material Co., LTD (Jinan City, China). Cupric sulfate (Analytical Reagent, AR), Tianjin Yongda Chemical Reagent Co (Tianjin City, China). Sodium borohydride (AR), Shanghai Shanpu Chemical Co (Shanghai City, China). Citric acid (AR), Tianjin Ruijin Chemical Reagent Co (Tianjin City, China). Ethylenediamine (AR), Shandong Yantai Laishan Economic Development Zone (Yantai City, China).

### 2.2. Synthesis of Carbon Nanofiber with Carboxymethyl Cellulose Hydrothermal Carbonization

60 mL H_2_O, and 2 g carboxymethyl cellulose, and 3 g citric acid, were added. Then, the above-mentioned mixture was transferred into a 100 mL Telfon-lined stainless-steel autoclave. The hydrothermal reaction was held at 200 °C for 3–6 h. The resulting product was washed with deionized water repeatedly 5–8 times, filtrated, 50 mL H_2_O was added, and ultrasound for 5–10 min. Polymer-derived carbon nanofiber suspension was obtained.

For comparison, carbon dots were synthesized with organic small molecules under similar conditions. 60 mL H_2_O, 20 mL ethylenediamine, and 3 g citric acid were added. Then, the above-mentioned mixture was transferred into a 100 mL Telfon-lined stainless-steel autoclave. The hydrothermal reaction was held at 200 °C for 3–6 h.

### 2.3. Defects Passivation of Carbon Nanofiber Derived with Carboxymethyl Cellulose with Cu Nanoparticles

5-, 10-, and 20 mL carbon nanofiber derived with carboxymethyl cellulose suspension, 200 mL H_2_O, were added, then 0.5 g cupric sulfate was added, respectively, followed by ultrasound for 5–10 min. Appropriate sodium borohydride was solved in 100 mL H_2_O, then added dropwise, respectively. After 24 h at room temperature, the resulting product was washed with deionized water repeatedly 5–8 times, filtrated, and 20mL H_2_O was added, followed by ultrasound for 5–10 min. Polymer-derived carbon nanofiber/Cu nanocomposite suspension was obtained.

### 2.4. Morphology Observation with SEM

The scanning electron microscopy (SEM) observation was carried out using a ZEISS Gemini SEM300 (Oberkohen City, Germany). The obtained sample was washed with deionized water, deposited on an aluminum foil substrate, dried at room temperature, and then sputtered with a thin layer of Pt on the surface for the SEM observation.

### 2.5. Energy Dispersive Spectroscopy (EDS) Measurements

Energy dispersive spectroscopy (EDS) measurements were performed using a Hitachi S-4800 (HITACHI, Tokyo, Japan). The obtained sample was washed with deionized water, deposited on Al foils, and dried at room temperature. The sample for EDS measurements did not need to have a layer of Pt deposited on its surface. The EDS mapping data were obtained.

### 2.6. Morphology Observation with TEM

The TEM observation was carried out using a JEM-1011 (Nippon Electronics Co., LTD. Showima City, Tokyo, Japan). The sample suspension was cast on copper mesh coated with carbon film, and dried at room temperature.

### 2.7. Measurement of UV-Vis-NIR Spectrum

The UV-Vis-NIR was determined using a TU-1810 spectrophotometer (Shanhai Yuan Analysis Instrument Co., LTD. Shanhai City, China) and the samples were obtained using suspension.

### 2.8. XRD Characterization

The powders’ X-ray diffraction (XRD) patterns were carried out using an XRD-7000 of SHIMADZU diffraction device (Shimadzu, Kyoto, Japan), a rotating anode X-ray generator working at 40 kV and 300 mA, with Cu Ka monochromatic radiation. The sample was dispersed in an aqueous solution, then cast on a glass substrate, and dried for 48–96 h at room temperature for determination.

### 2.9. Photocurrent Response of Nanocomposite to Visible Light and NIR

The resulting nanocomposite suspension was cast on the Au gap electrodes on a flexible PET (polyethylene terephthalate) film substrate. After drying, the photoconductive response to weak visible light (40 W) or 650 nm (100, 50, and 5 mW) and 808, 980, and 1064 nm NIR with low-power (10, 40, 50, 100, and 200 mW) was determined using an LK2000A Electrochemical Work Station from LANLIKE Chemistry and Electron High Technology Co., Ltd. (Tianjin City, China), applied 1V or -1V DC bias. The current of the thick film was determined by computer recording before and after the irradiation of light.

## 3. Results and Discussion

According to previous reports of references, after dehydration and carbonization of the small molecules or polymer precursors, graphene quantum dots (GQDs), carbon quantum dots (CQDs), carbon nanodots (CNDs), and carbonized polymer dots (CPDs) would be obtained. In our experiments, some similar experiments with several organic small molecules (such as cyclodextrin, glucose, urea, thiourea, ferrocene, cysteine, glutamic acid, serine, etc.) and polymer precursors (such as chitosan, polyvinyl alcohol, polyacrylamide, sodium alginate, starch, protein, leaf, etc.) were also carried out. Different morphologies of carbon nanostructures were obtained. Some are carbon dots, and others are nanofibers or nanosheets, etc. Since carboxymethyl cellulose has some carboxyl groups, and a large number of hydroxyl groups, to increase the degree of polycondensation and crosslinking for polymer clusters, in the process of carbonization of carboxymethyl cellulose, some amount of citric acid was added. For comparison, carbon dots were synthesized with citric acid small molecules under similar conditions. The results indicated that carbonization of citric acid, and carbon dots were obtained (Figure 1a), and for carbonization of crosslinked carboxymethyl cellulose, carbon nanofiber was gained (Figure 1b). The representative TEM image of carbon dots and nanofiber are shown in Figure 1a,b.

As shown in Figure 1a, the carbon dots synthesized with citric acid showed nanoparticle structure. The diameter of these particles is in the range of 20–80 nm. And as shown in Figure 1b, carbon nanofiber was obtained by carbonization of crosslinked carboxymethyl cellulose. The diameter of these nanofibers is about 20 nm or so, and the length is in the range of 50–100 nm. It exhibited an interconnected dendritic-like structure. Due to carboxymethyl cellulose possessing some carboxyl groups and a large number of hydroxyl groups, the citric acid molecule containing three carboxyl groups and one hydroxyl group; in the process of carbonization of carboxymethyl cellulose, polycondensation between carboxyl and hydroxyl groups occurred. Citric acid acted as the role of crosslinking agent, enhancing the crosslinking degree of polymer. The condensation between the carboxyl groups of carboxymethyl cellulose and the hydroxyl groups possibly took place in inter- or intra-polymer chains, and led to the formation of an interconnected dendritic-like structure.

The representative SEM image of polymer-derived carbon/Cu nanocomposite is shown in Figure 2.

As shown in Figure 2, most of the sample region was covered by nanoparticles, these nanoparticles belong to Cu, and their size is about 20–80 nm. Some local areas (Figure 2) showed a nanofiber-like structure. These interconnected dendritic-like structures are polymer-derived carbon nanomaterials, and are covered by Cu nanoparticles. This close contact is contributing to interface charge transfer between polymer-derived carbon nanofiber and Cu nanoparticles.

The XRD results of polymer-derived carbon nanofiber/Cu nanocomposite are shown in Figure 3. The UV-Vis-NIR of polymer-derived carbon nanofiber/Cu nanocomposite is shown in Figure 4.

As shown in Figure 3, the diffraction peaks at 40.39°, 50.51°, and 74.84° are the peaks of (111), (200), (220) planes of Cu (PDF# 04-0836), respectively. Otherwise, the diffraction peaks at 20.49°, 26.25°, 42.44°, and 44.87° are the peaks of (101), (002), (100), (101) planes of graphite (PDF# 41-1487), respectively. Therefore, the resulting nanocomposite contains Cu and carbon of graphite.

Figure 4 shows that the absorbance of polymer-derived carbon nanofiber/Cu nanocomposite covered the whole visible light range, and widened to the NIR. The absorbance band edge is bigger to 1100 nm (the instrumental limit used). It is expected that the resulting nanocomposite had good potential applications for photodetectors, bio-imaging, biomedical field, information science, etc. Regarding photoelectric functional materials, it is well-known that transient fluorescence, transient absorption spectroscopy, and transient photoconductivity are widely used to study the mechanism of recombination of photo-induced charge generation and determine the lifetime of electrons/holes generated in many published references. The lifetime is generally located in ps order (10^−10^–10^−13^s). However, free electron/hole generation would be needed for a long timescale for many potential applications. Free electrons/holes with longer lifetimes would participate in a series of oxidation and reduction reactions, which are favored for applications in energy and environmental fields, especially for the degradation of organic pollutants or some typical reduction reactions. Therefore, the photoconductive response to weak visible light was studied based on Au gap electrodes on flexible PET (polyethylene terephthalate) film substrate by casting thick film in a long timescale. The typical results are shown in Figure 5.

Figure 5 shows that the comparative photocurrent responses of polymer–derived carbon nanofiber/Cu nanocomposite with different content of carbon nanofiber to weak visible light. The horizontal axis of Figure 5 is the response time (s). The vertical axis of Figure 5 is the value of film current (a: the black and red line represented the different loading of carbon nanofiber is 5, 20 mL, respectively; b: the loading of carbon nanofiber is 10 mL in Figure 5). As shown in Figure 5, when the resulting nanocomposite thick film was exposed to weak visible light, the current flowing through the film increased dramatically due to the carrier’s photo-generated separation. On the contrary, the film current decreased greatly when the visible light was off. The response time is about 37.05, 15.03 s, the recovery time is about 37.04, 29.53 s, and the ratio of On/Off is about 2.61, 5.51 for the nanocomposites loaded carbon nanofiber (5, 20 mL). This illustrates that the resulting nanocomposite thick film produced photo-induced charges easily to weak visible light, possessing good visible light activities. Therefore, free electrons/holes can be generated and separated with the excitation of the low-power visible light resources. It is interesting that when the loading carbon nanofiber is 5, 20 mL, the polymer-derived carbon nanofiber/Cu nanocomposite showed positive photoconductance responses, and while the loading carbon nanofiber is 10 mL, the resulting nanocomposite exhibited negative photoconductance responses. The response time is about 37.05 s, the recovery time is about 29.54 s, and the average ratio of On/Off is about 2.56. The baseline current is also a great fluctuation. This behavior is contributing to electron trap-assisted photocurrent response. A large number of electrons generated by light inducement were trapped by the charged defects of the nanocomposites, holes were collected at the interface between polymer-derived carbon nanofiber and Cu nanoparticles, the direction of built-in electrical fields produced is opposite to that of bias fields applied, and resulted in the current decreasing.

Regarding photoactive functional materials in the visible light range, besides potential applications in energy and environmental fields, simulating the retina is also important in biomedical fields. Especially, 532, 650 nm light response film is often used for the artificial retina or repair vision. Photoelectric imaging simulated vision. Therefore, 100 mW 650 nm was selected as typical light resources to examine photocurrent responses of polymer-derived carbon nanofiber/Cu nanocomposite with different content of carbon nanofiber. The results are shown in Figure 6.

Figure 6 shows that the comparative photocurrent responses of polymer–derived carbon nanofiber/Cu nanocomposite with different content of carbon nanofiber to 100 mW 650 nm light. As shown in Figure 6 (the black, red, and blue lines represented the different content of carbon nanofiber loading is 5, 10, and 20 mL, respectively in Figure 6), when the resulting nanocomposite thick film was exposed to 100 mW 650 nm light, the current flowing through the film increased dramatically due to the carrier’s photo-generated separation. On the contrary, the film current decreased greatly when the light resource was off. The response time is about 22.02, 37.05, 22.02 s, the recovery time is about 14.49, 15.03, 7.52 s, and the ratio of On/Off is about 21.65, 6.97, 8.05 for the nanocomposites loaded carbon nanofiber (5, 10, 20 mL). This illustrates that the resulting nanocomposite thick film produced photo-induced charges easily to 100 mW 650 nm light. Therefore, free electrons/holes can be generated and separated with the excitation of the 100 mW 650 nm light resources.

In the biomedical or interdisciplinary fields, 808, 980, and 1064 nm are also vital light resources in the applications of cancer therapy, bio-imaging, artificial vision, remoting drug delivery by light, photodetector, etc. The photocurrent responses of polymer-derived carbon nanofiber/Cu nanocomposite to 808, 980, and 1064 nm light were also examined. The results are shown in Figure 7, Figure 8 and Figure 9.

Figure 7 shows that the comparative photocurrent responses of polymer-derived carbon nanofiber/Cu nanocomposite with different content of carbon nanofiber to 200 mW 808 nm. As shown in Figure 7 (the black, red, and blue line represented the different content of carbon nanofiber loading is 5, 10, and 20 mL, respectively in Figure 7), when the resulting nanocomposite thick film was exposed to 200 mW 808 nm light, the current flowing through the film increased dramatically due to the carrier’s photogenerated separation. On the contrary, the film current decreased greatly when the light resource was off. The response time is about 15.03, 66.58, 7.52 s, the recovery time is about 22.01, 22.01, 15.03 s, and the ratio of On/Off is about 11.52, 3.15, 22.28 for the nanocomposites loaded carbon nanofiber (5, 10, 20 mL). Otherwise, it is found that the base current of the thick film decreased with increasing the contents of carbon nanofiber. This illustrates that the conductivity of polymer-derived carbon nanofiber is poor than that of Cu, and the resulting nanocomposite thick film produced photo-induced charges easily to 200 mW 808 nm light. Therefore, free electrons/holes can be generated and separated with the excitation of the 200 mW 808 nm light resources.

Figure 8 showed that the comparative photocurrent responses of polymer-derived carbon nanofiber/Cu nanocomposite with different content of carbon nanofiber to 200 mW 980 nm. As shown in Figure 8 (the red, blue, and green line represented the different content of carbon nanofiber loading is 5, 10, and 20 mL, respectively in Figure 8), when the resulting nanocomposite thick film was exposed to 200 mW 980 nm light, the current flowing through the film increased dramatically due to the carrier’s photo-generated separation. On the contrary, the film current decreased greatly when the light resource was off. The response time is about 15.03, 15.038, 29.53 s, the recovery time is about 29.53, 2.77, 13.79 s, and the ratio of On/Off is about 11.52, 3.15, 22.28 for the nanocomposites loaded carbon nanofiber (5, 10, 20 mL). When the nanocomposites loaded carbon nanofiber is 5, 20 mL, the results showed good photoelectric responses, and when the loading of carbon nanofiber is 10 mL, the baseline current showed a great fluctuation. This illustrates that the interface contact was different with the changing composition ratio of polymer-derived carbon nanofiber/Cu, and led to different carriers extracting abilities; the resulting nanocomposite thick film produced photo-induced charges easily to 200 mW 980 nm light. Therefore, free electrons/holes can be generated and separated with the excitation of the 200 mW 980 nm light resources.

Figure 9 shows that the comparative photocurrent responses of polymer–derived carbon nanofiber/Cu nanocomposite with different content of carbon nanofiber to 40 mW 1064 nm. As shown in Figure 9 (a: the black and red lines represented the different loading of carbon nanofiber is 5, 20 mL; b: the loading of carbon nanofiber is 10 mL), when the resulting nanocomposite thick film was exposed to 40 mW 1064 nm light resource, the current flowing through the film increased dramatically due to the carrier’s photo-generated separation. On the contrary, the film current decreased greatly when the light resource was off. The response time is about 29.53, 22.55 s, the recovery time is about 51.55, 15.04 s, and the ratio of On/Off is about 1.64, 1.33 for the nanocomposites loaded carbon nanofiber (5, 20 mL). This illustrates that the resulting nanocomposite thick film produced photo-induced charges easily to 40 mW 1064 nm light resource. Therefore, free electrons/holes can be generated and separated with the excitation of the low-power 1064 nm NIR light resources. It is interesting that when the loading of carbon nanofiber is 5, 20 mL, the polymer-derived carbon nanofiber/Cu nanocomposite showed positive photoconductance responses, and while the loading of carbon nanofiber is 10 mL, the nanocomposite exhibited negative photoconductance responses. The response time is about 30.07 s, the recovery time is about 52.09 s, and the average ratio of On/Off is about 7.87. This behavior is also contributing to electron trap-assisted photocurrent response. A large number of electrons generated by light inducement were trapped by the charged defects of the nanocomposites, holes were collected at the interface between polymer-derived carbon nanofiber and Cu nanoparticles, the direction of built-in electrical fields produced is opposite to that of bias fields applied, and resulted in the current decreasing. This behavior is similar to that of the visible light response. Otherwise, regarding the nanocomposite-loaded carbon nanofiber (5 mL), in the beginning, the baseline current was stabilized. However, after several cycles of on/off, the baseline current changed with a great fluctuation. It illustrated that the stability of polymer-derived carbon nanofiber (5 mL)/Cu nanocomposite was poor than that of polymer-derived carbon nanofiber (20 mL)/Cu. This is the result of the unbalance of carriers trapped and detrapped, which involved microstructure, defects, interfaces, and photo-doping effects of the nanocomposite.

In short, as mentioned above these results, the polymer-derived carbon nanofiber/Cu nanocomposite with different content of carbon nanofiber showed different photocurrent responses by changing excitation wavelength from visible range to NIR. For 650, 808, and 980 nm light resources, the above three nanocomposites with changing composition ratios exhibited good photocurrent-switching responses. Regarding the nanocomposites with different composition ratios and changing excitation light wavelengths (such as 1064 nm, visible light range), different photophysical mechanisms led to different photocurrent responses. This is mainly responding to the effects of the microstructure, defects, and interfaces of nanocomposite on the contribution of the carrier’s photo-generated extracting ability. Herein, photocurrent responses were examined with lower power of typical light resources and representative nanocomposites. The results are shown in Figure 10 and Figure 11.

Figure 10 shows the comparative photocurrent responses of polymer-derived carbon nanofiber/Cu nanocomposite with different contents of carbon nanofiber to 100, 50, and 5 mW 650 nm. Figure 11 shows the comparative photocurrent responses of polymer-derived carbon nanofiber/Cu nanocomposite with different contents of carbon nanofiber to 200, 100, 50, and 5 mW 980 nm. As shown in Figure 10 (where the black and red lines represent the different loadings of carbon fiber: 5, 20 mL, and 100, 50, and 5 mW of 650 nm), and Figure 11 (a, where the red and blue lines represent the different loadings of carbon nanofiber is 5, 20 mL to 200, 100, 50, and 5 mW of 980 nm; b, partially enlarged to 5 mW for the loading of carbon fiber at 20 mL in Figure 11), it is found that the nanocomposites loaded carbon nanofiber (5, 20 mL) still had good photocurrent responses to 50 mW 650 nm of light. However, there is little response to 5 mW 650 nm of light. Similarly, the nanocomposites loaded with carbon nanofiber (20 mL) still had good photocurrent responses to 5 mW 980 nm of light. The partially enlarged Figure to 5 mW for the loading of carbon fiber (20 mL) is shown in Figure 11b. This illustrated that the polymer-derived carbon nanofiber (20 mL)/Cu nanocomposite was more sensitive than that of other nanocomposites with carbon nanofiber loading in NIR.

To explore the effects of the contents of loading of carbon nanofiber on the carrier’s photo-generated extracting ability, the energy dispersive spectroscopy (EDS) measurements were performed. The representative results of carbon EDS mapping data, copper EDS mapping data, and oxygen EDS mapping data of the nanocomposites are shown in Figure 12, Figure 13 and Figure 14.

Figure 12, Figure 13 and Figure 14 show the representative results of carbon EDS mapping data, copper EDS mapping data, and oxygen EDS mapping data of the resulting nanocomposite (a, b, and c represented the different content of carbon nanofiber loading in the nanocomposites is 5, 10, and 20 mL, respectively in Figure 12, Figure 13 and Figure 14).

As shown in Figure 12, Figure 13 and Figure 14, it can be found that the distribution of carbon, copper, and oxygen element of nanocomposites is uniform as a whole. However, the aggregation of carbon nanofiber was observed in a few local areas. Otherwise, with increasing the contents of the carbon nanofiber in nanocomposite loading, the contents of the oxygen element were decreased, which would be in favor of improving the oxidation resistance of copper nanoparticles [58,59,60]. This could provide further indirect evidence to explore the interface interaction between carbon nanofiber and Cu nanoparticles.

## 4. Conclusions

In summary, carbon nanofiber was obtained via polymer hydrothermal carbonization. Based on the plasmonic characteristics and rich electrons in the Cu nanostructure, the Cu nanoparticles acted in the role of the electronic donor in passivating defects within the carbon nanofiber, improving the carrier’s photo-generated extracting ability. The resulting nanocomposite solid thick film showed the broadband spectrum photocurrent-switching behaviors from visible light to NIR, the switch ratio was dependent on the wavelengths and power of incident light. The positive and negative photoconductance responses phenomenon was observed in changing composition of nanocomposites and different excited wavelengths. Their mechanisms mainly contributed to the charged defects-assisted charge transfer of nanocomposites. This illustrated that the nanocomposites easily produce free electrons and holes via low power of incident light. Free electrons and holes would be utilized for different purposes in multi-disciplinary fields. These results are in favor of exploring the mechanism of light/matter interaction. It would be a potential application in the broadband flexible photodetector, cancer therapy, bio-imaging, artificial vision, remoting drug delivery via light or environmental fields, such as photocatalysts, and treatment of organic pollutants from visible light to NIR. This is a low-cost and green approach to obtaining nanocomposites exhibiting good photocurrent response from the visible light range to NIR.

Several conclusions are as follows: (1) carbon nanofiber was synthesized with crosslinked carboxymethyl cellulose via hydrothermal carbonization. (2) the strong interface interaction between carbon nanofiber and Cu nanoparticles enhanced the carrier’s photo-generated extracting ability, which showed the broadband spectrum photocurrent-switching responses from visible light to NIR. (3) The positive and negative photoconductance responses phenomenon was observed in changing composition of nanocomposites and different excited wavelengths, which correspond to different physical mechanisms. These results are in favor of exploring the mechanism of light/matter interaction.

## Figures and Tables

**Figure 1 polymers-15-03528-f001:**
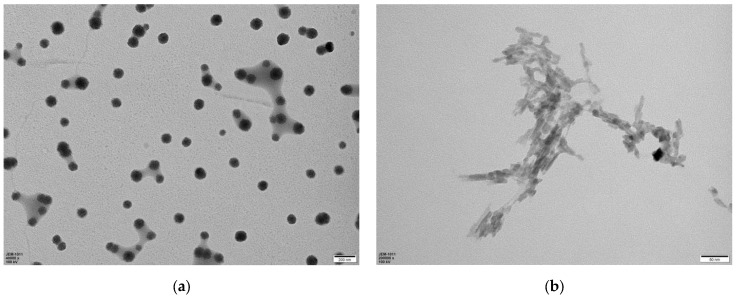
The representative TEM image of carbon dots and nanofiber (**a**): citric acid-derived carbon dots; (**b**): polymer-derived carbon nanofiber.

**Figure 2 polymers-15-03528-f002:**
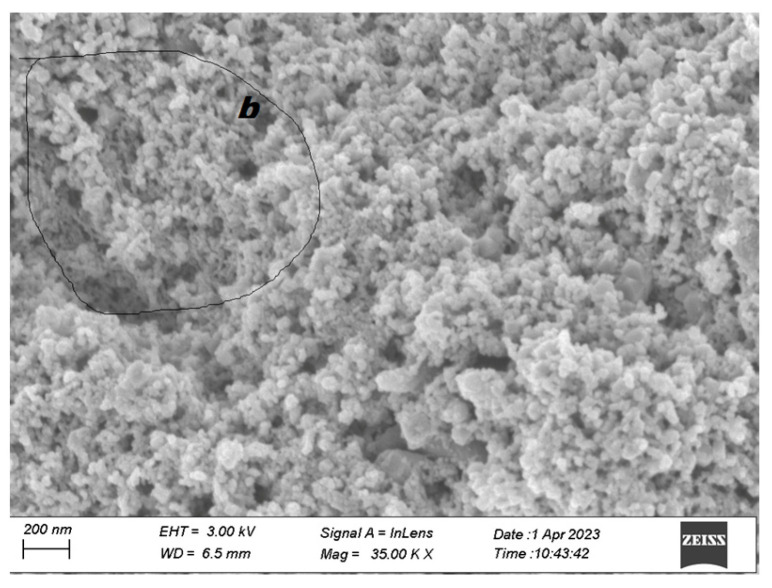
The representative SEM image of polymer-derived carbon/Cu nanocomposite: representative SEM; b: local area of SEM.

**Figure 3 polymers-15-03528-f003:**
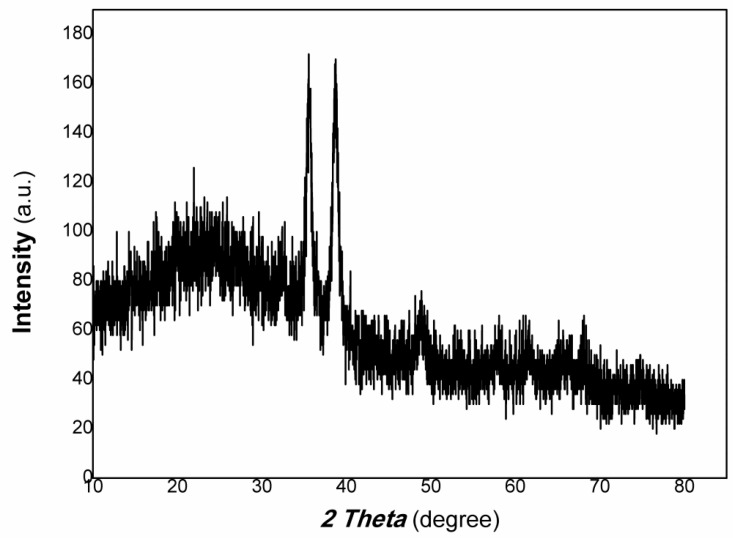
The XRD results of polymer-derived carbon nanofiber/Cu nanocomposite.

**Figure 4 polymers-15-03528-f004:**
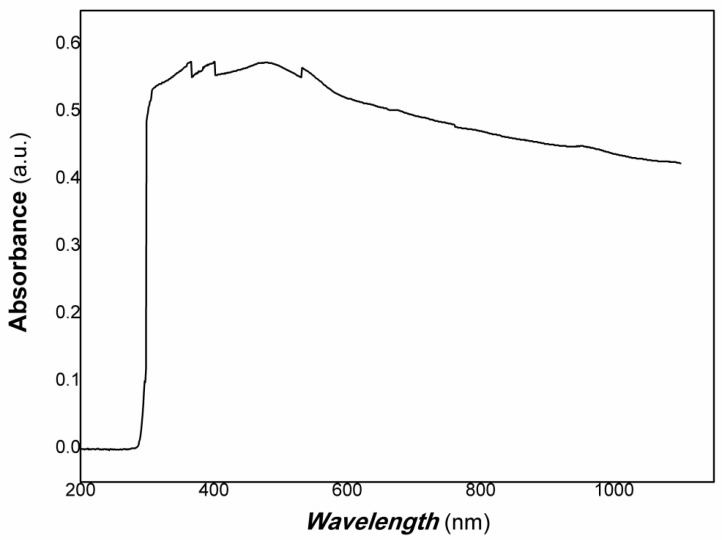
The UV-Vis-NIR of polymer-derived carbon nanofiber/Cu nanocomposite.

**Figure 5 polymers-15-03528-f005:**
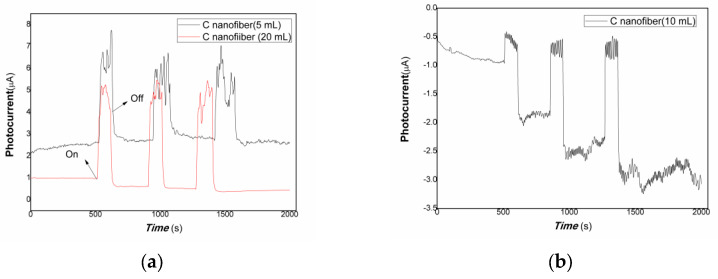
The comparative photocurrent responses of the resulting nanocomposites to weak visible light (**a**): the black and red line represented the different loading of carbon nanofiber is 5, 20 mL; (**b**): the loading of carbon nanofiber is 10 mL).

**Figure 6 polymers-15-03528-f006:**
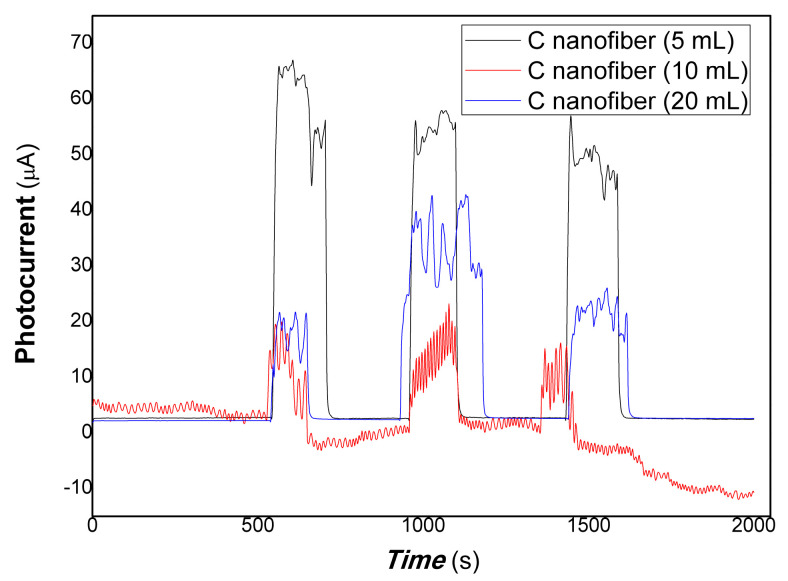
The comparative photocurrent responses of the resulting nanocomposites to 100 mW 650 nm light.

**Figure 7 polymers-15-03528-f007:**
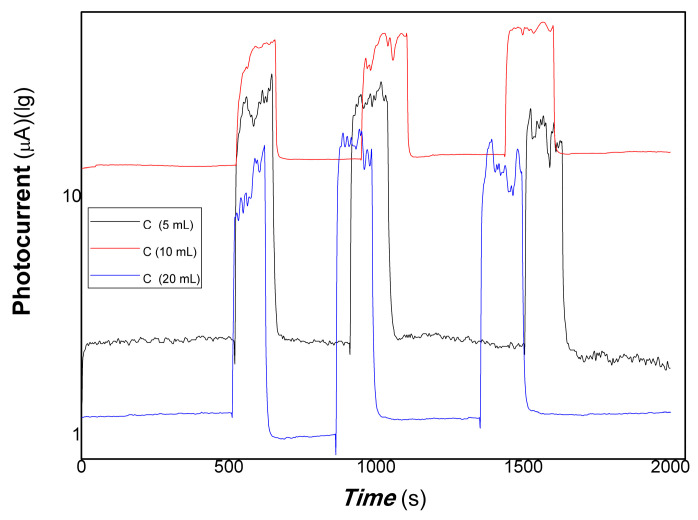
The comparative photocurrent responses of the resulting nanocomposites to 200 mW 808 nm.

**Figure 8 polymers-15-03528-f008:**
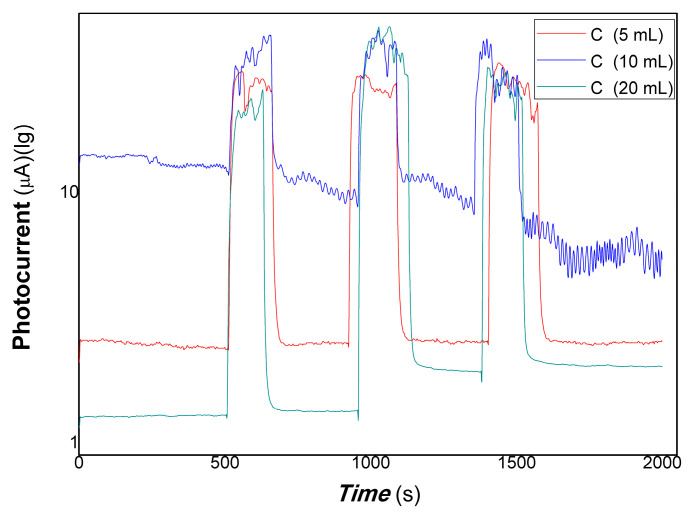
The comparative photocurrent responses of the resulting nanocomposites to 200 mW 980 nm.

**Figure 9 polymers-15-03528-f009:**
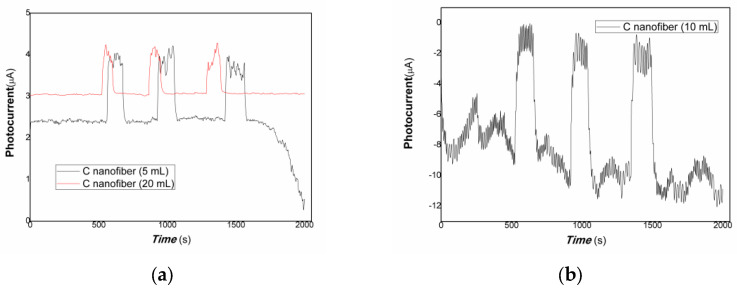
The comparative photocurrent responses of the resulting nanocomposites to 40 mW 1064 nm (**a**): the black and red lines represented the different loading of carbon nanofiber is 5, 20 mL; (**b**): the loading of carbon nanofiber is 10 mL).

**Figure 10 polymers-15-03528-f010:**
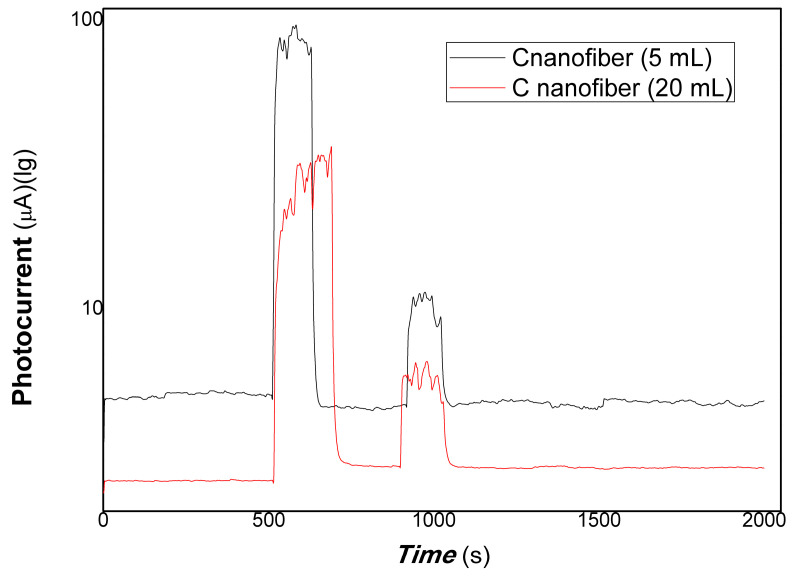
The comparative photocurrent responses of the resulting nanocomposites to 100, 50, 5 mW 650 nm.

**Figure 11 polymers-15-03528-f011:**
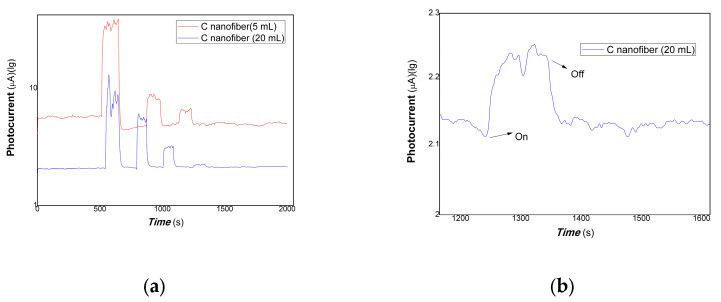
The comparative photocurrent responses of the resulting nanocomposites to 200, 100, 50, 5 mW 980 nm (**a**), where the red and blue lines represent the different loadings of carbon nanofiber is 5, 20 mL to 200, 100, 50, and 5 mW of 980 nm; (**b**), partially enlarged to 5 mW for the loading of carbon fiber at 20 mL).

**Figure 12 polymers-15-03528-f012:**
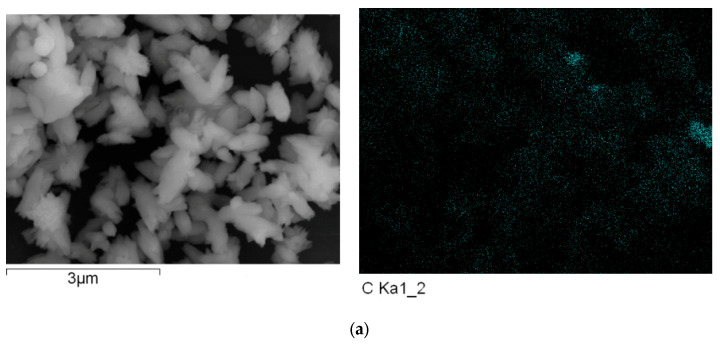
The energy dispersive spectroscopy (EDS) of carbon EDS mapping of the resulting nanocomposite with different content of carbon nanofiber. (**a**) a-Carbon EDS mapping; (**b**) b-Carbon EDS mapping; (**c**) c-Carbon EDS mapping.

**Figure 13 polymers-15-03528-f013:**
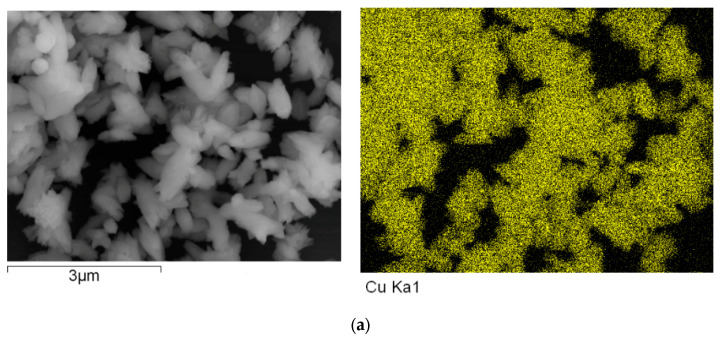
The energy dispersive spectroscopy (EDS) of copper EDS mapping of the resulting nanocomposite with different content of carbon nanofiber. (**a**) a-Copper EDS mapping; (**b**) b-Copper EDS mapping; (**c**) c-Copper EDS mapping.

**Figure 14 polymers-15-03528-f014:**
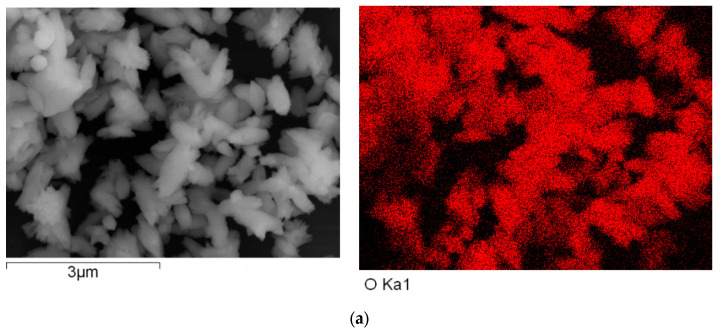
The energy dispersive spectroscopy (EDS) of oxygen EDS mapping of the resulting nanocomposite with different content of carbon nanofiber. (**a**) a-Oxygen EDS mapping; (**b**) (b-Oxygen EDS mapping; (**c**) c-Oxygen EDS mapping.

## Data Availability

The data presented in this study are available on request from the corresponding author. The data are not publicly available due to privacy.

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
