# Peer review of "Polymer-Derived Carbon Nanofiber and Its Photocurrent-Switching Responses of Carbon Nanofiber/Cu Nanocomposite in Wide Ranges of Excited Light Wavelength"

_polymers, 2023, doi:10.3390/polym15173528_

Round 1

Reviewer 1 Report

Please see the attched file.

Author Response

Dear Sir, 

Thank you for your work on our paper. The response to comments have been submited it with PDF file.

Sincerely,

Xingfa Ma

Reviewer 2 Report

This manuscript reports the carbon nanofiber/Cu nanocomposite as photocurrent-switching responses in wide light wavelength ranges. Similar previous papers (such as ACS Sustainable Chem. Eng. 10, 38, 12630–12641 (2022); Composites Communications 34, 101256 (2022), Journal of Electronic Materials, 49, 3165–3173 (2020) have already been discussed. So why is the current structure sufficiently new or exciting to warrant publication? Although some results seem interesting, after careful reading of this work, I cannot recommend publication in its present form.

1.        Why this work merits publication? I cannot find significant advances with those reported previously since the carbon nanofiber Cu nanocomposite has been studied intensively for the past few years. The authors should strengthen the novelty and differences of this work in the introduction part.

2.        The presentation of the data chart is very rough, much like the original data.

3.        The authors should provide the XPS and FTIR data of carbon nanofiber/Cu nanocomposite.

4.        The authors should provide the TEM and TEM-EDS mapping data of carbon nanofiber/Cu nanocomposite.

5.        There are several mistakes. Authors should check for spelling, symbols that should be given in Italics, space between words and numbers etc.

6.        Some of the words in the figures are not clear. If the authors can modify it would be better.

Moderate editing of English language required.

Author Response

Dear Sir

Thank you for your work on our paper. The response to your comments have been submitted with PDF file.

Sincerely,

Xingfa Ma

Round 2

Reviewer 1 Report

The manuscript can in principle be accepted.

1. The figures (Fig. 5 to Fig. 11) are mixed, too long, with some poor resolution images some differences in size, and cannot be followed clearly by the readers.

2. BSEs (backscattered electrons), and EDS are not provided to study the chemical and physical properties of the samples (Cu/carbon). BSEs can be used to obtain high-resolution images that show the distribution of various elements that make up the C samples with different Cu additions.

3. The details of the obtained carbon sample identification by the XRD diffraction is not provided. What is the software, and the database being used?

4. Results and Discussion is one long section.

Author Response

Comments-1

Dear Sir,

Thank you for your comments. We have revised it in the revision.

Sincerely,

Xingfa Ma

The manuscript can in principle be accepted.

  1. The figures (Fig. 5 to Fig. 11) are mixed, too long, with some poor resolution images some differences in size, and cannot be followed clearly by the readers. 

Thank you very much! We revised it a little in the revision.

  1. BSEs (backscattered electrons), and EDS are not provided to study the chemical and physical properties of the samples (Cu/carbon). BSEs can be used to obtain high-resolution images that show the distribution of various elements that make up the C samples with different Cu additions.

Thank you very much! The EDS mapping data has been added. Now, it is in summer holiday, the EDS experiments were carried out with another SEM (S-4800). It is shown in Line 228-233, 499-525. The carbon EDS mapping data, copper EDS mapping data, and oxygen EDS mapping data of the resulting nanocomposite (the different content of carbon nanofiber loading in nanocomposites is 5, 10, and 20 mL respectively) are shown in Fig. 12-14.

  1. The details of the obtained carbon sample identification by the XRD diffraction is not provided. What is the software, and the database being used?

 Thank you very much! Based on some references of polymer-derived carbon dots, it is difficult to obtain the XRD diffraction due to polymer nanodot formed from a polymer with no lattice structure. We have no carbon sample identification by the XRD diffraction. This time, special Thanks for your warning again. The diffraction peaks at 20.49°, 26.25°, 42.44°, and 44.87° can be distinguished, which are the peaks of (101), (002), (100), (101) planes of graphite (PDF# 41-1487) respectively, although the diffraction peaks are a little low. It is shown in Line 304-306.

  1. Results and Discussion is one long section.

Thank you very much again! Yes. This study is focused on the broadband spectrum (from the visible light region to NIR) range. In the visible light region, 650 nm was selected; In the NIR, 808, 980, 1064 nm were selected. In other relative publications, most of research is put on 1 or 2 wavelength. Therefore, it appears a little long. It doesn't compress the length of paper very well.

Reviewer 2 Report

The authors have revised the manuscript according to the comments. Therefore, I recommend that the manuscript could be accepted for publication.

Minor editing of English language required.

Author Response

Comments-2

The authors have revised the manuscript according to the comments. Therefore, I recommend that the manuscript could be accepted for publication.

Dear Sir,

Thank you for your comments. We have checked the English again. Due to not professional, some neglectful issue is possible presence.

Sincerely,

Xingfa Ma